# Investigation on the Dynamic Behavior of Weld Pool and Weld Microstructure during DP-GMAW for Austenitic Stainless Steel

**Tao Chen [1], Songbai Xue [1,\*] , Peng Zhang [1], Bo Wang [2], Peizhuo Zhai [1] and Weimin Long [2]**

[1] College of Materials Science and Technology, Nanjing University of Aeronautics and Astronautics, Nanjing 210016, China; taocmsc@nuaa.edu.cn (T.C.); mstzhangpeng@nuaa.edu.cn (P.Z.); zhaipz@nuaa.edu.cn (P.Z.)

[2] Institute of Advanced Brazing Materials and Technology, China Intelligent Equipment Innovation Institute (Ningbo) Co., Ltd., Ningbo 315700, China; wangbo4175@126.com (B.W.); brazelong@163.com (W.L.)

\* Correspondence: xuesb@nuaa.edu.cn; Tel.: +86-8489-6070

**Abstract:** The influence of heat and droplet transfer into weld pool dynamic behavior and weld metal microstructure in double-pulsed gas metal arc welding (DP-GMAW) was investigated by the self-designed high-speed welding photography system. The heat input, the arc pressure, the droplet momentum and impingement pressure were measured and calculated. It was found that the arc pressure is far less than the droplet impingement pressure. The heat input and droplet impingement pressure per unit time acting on weld pool were proportional to the current pulse frequency, which fluctuated with thermal pulse. The size and oscillation amplitude of the weld pool had noticeable periodic changes synchronized with the process of heat input and droplet impingement. Compared to the microstructure of pulsed gas metal arc welding (P-GMAW) weld metal, that of DP-GMAW weld metal was significantly refined. High oscillation amplitude assisted the enhancement of weld pool convection, which leads to more constitutional supercooling. The heat input and shear force during the peak of thermal pulse causing dendrite fragmentation which provided sufficient crystal nucleus for the growth of equiaxed grains and the possibility of grain refinement. The effects of current parameters on welding behavior and weld metal grain size are investigated for further understanding of DP-GMAW.

**Keywords:** double-pulsed gas metal arc welding (DP-GMAW); droplet impingement pressure; weld pool oscillation; grain refinement; constitutional supercooling

---

## 1. Introduction

As a widely used spatter-free welding technology, pulsed gas metal arc welding (P-GMAW) can achieve directional transition of spatter-free droplets with low heat input through current pulse [1]. However, current pulses with the constant frequency of P-GMAW could not effectively stir the weld pool with the heat-sensitive and high viscosity liquid metal such as stainless steel and aluminum alloys, which often results in the formation of structure defects such as coarse grains, pores and cracks [2]. To solve this problem, many arc-based welding techniques have been developed for advanced materials joining [3,4]. DP-GMAW was developed based on P-GMAW to assisting weld pool oscillation [5]. By periodically changing the output current, double-pulsed gas metal arc welding (DP-GMAW) leads to the periodic change of current pulse frequency. Through the whole process, not only the stable transfer mode of "one drop per pulse" can be obtained [6], but also the frequency oscillation and stirring effect of weld pool can be obviously improved [7]. Therefore, to some extent, DP-GMAW

can refine the grains [2], reduce the cracking sensitivity [8,9] and porosity of welds [3,4] and improve the weld formation and joint performance.

Figure 1 shows the waveform of DP-GMAW, and the double pulse period consists of a peak thermal period and a base thermal period. The heat and mass transfer process of the heat pulse is determined by the waveform parameters of double-pulse current such as pulse frequency, current difference and duty ratio of two phases. To study the evolution rules of DP-GMAW weld formation and microstructure, various methods were carried out. Yao and Zhou et al. systematically investigated the influence of current waveform parameters of DP-GMAW on the weld waviness of austenitic stainless steel [10]. They discussed the regularity of DP-GMAW weld formation by using gray theory analysis [11] and further optimized the welding parameters of austenitic stainless steel [12]. Compared with P-GMAW, DP-GMAW has a wider adjustment range, broader root gap configuration and stronger solute agitation with the same heat input rate. Wang et al., suggested that increasing the frequency of thermal pulse of DP-GMAW could reduce dendrite size [13]. Wang studied the influence of the current amplitude of thermal pulse on the geometry, cooling rate, solidification parameters of the aluminum alloy weld pool and weld metal grain size from both experimental and numerical simulation aspects. It was proved that DP-GMAW could increase the cooling rate of the weld pool with the same heat input [14]. In the investigation on welding procedure of ferritic stainless steel and austenitic stainless steel, Shen [8] and Devakumaran [9] both confirmed that DP-GMAW could effectively inhibit the growth of HAZ (Heat Affected Zone) grains and promote the transformation of columnar grains to equiaxed grains in the weld zone. Anhua Liu et al. analyzed the dynamic process of weld pool shape of the aluminum alloy with the aid of high-speed camera [15]. The results showed that the size of weld pool changed synchronously with the frequency the thermal pulse, and the addition of thermal pulse obviously changed the behaviors of weld pool. As the frequency of thermal pulse increased, the grain size of weld metal decreased, and the eutectic $Mg_2Si$ precipitates in the weld zone were evenly distributed.

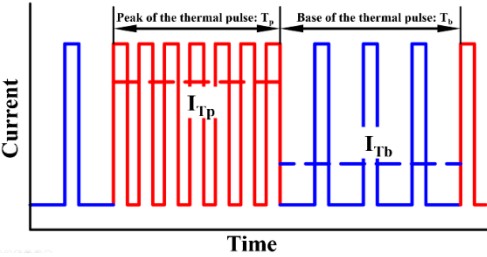

**Figure 1.** Schematic diagram of current waveform of double-pulsed gas metal arc welding (DP-GMAW).

It should be pointed out that the heat and mass transfer process during welding determines the dynamic behavior of the weld pool. Besides, the solidification behavior of weld pool is greatly controlled by the dynamic behavior of weld pool and welding heat input process. So far, studies on the effect of DP-GMAW on grain refinement of the weld mainly concentrated on the optimizing welding procedure. However, little research was conducted on the dynamic characteristics of weld pool in DP-GMAW, and the relationship between the dynamic behavior of weld pool and welding metal microstructure can hardly be established. Therefore, extensive research work needs to be carried out to analyze the influence of waveform parameters of DP-GMAW on the dynamic behavior of weld pool and its relationship with welding metal microstructure.

In this paper, with the help of a laboratory-made high-speed welding photography system, the influence of waveform parameters of DP-GMAW on weld pool oscillation behavior of austenitic stainless steel was studied. The purpose of this study is to explore the internal relationship between weld pool oscillation behavior and welding metal microstructure and explain the action mechanism of grain refinement.

## 2. Materials and Methods

### 2.1. Experiment System

The experimental system used in this study consisted of a welding system, high-speed photographic system and an image processing system, as shown in Figure 2. An 850 nm laser source and some 850 nm near-infrared filters equipped on camera were used in a high-speed photographic system to eliminate the light of strong arc. A high-speed camera was located in different positions to obtain weld pool images with various visual angles. The side-view of the weld pool during welding was recorded in position 1, as shown at the position of high-speed camera 1 in Figure 2. The top-view of the weld pool was recorded in position 2, as shown at the position of highspeed camera 2 in Figure 2. Otto Arc MIG-500DP (OTTO Arc, Shanghai, China) was selected as a welding power source. Welding position was PA (Flat position, as per ISO 6947). The frequency of an electric signal acquisition system was $5 \times 105$ Hz.

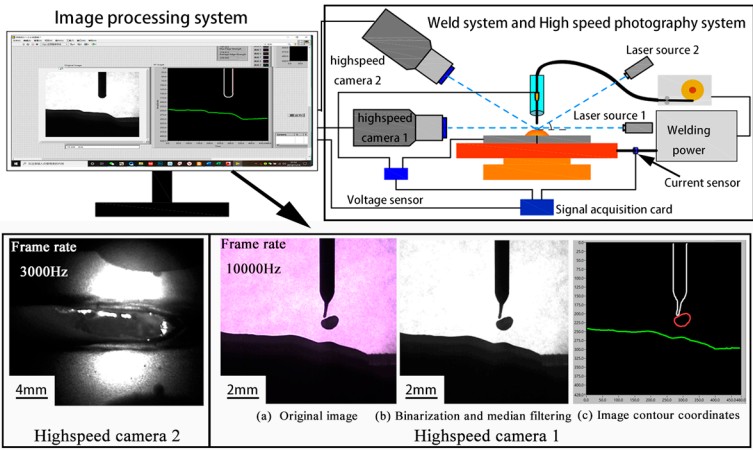

**Figure 2.** Experiment system and image processing steps: (**a**) original image; (**b**) binarization and median filtering; (**c**) image contour coordinates.

An image processing system based on LabView (LabVIEW 2017, National Instruments, Austin, TX, USA) was developed to capture the outline of droplet and pool surface from the side-view picture. The image processing flow is shown in Figure 2a–c.

### 2.2. Algorithm to Extract Characteristics of Pool Oscillation, Droplet Transfer and Arc Profile

The outlines of droplet and pool surface captured by the image processing system are shown in Figure 2c. The contour coordinates were deformed with the droplet transfer and the pool oscillation. The dynamic information of the droplet and weld pool can be obtained by tracing the contour coordinates as a function of time, as shown in Figure 3.

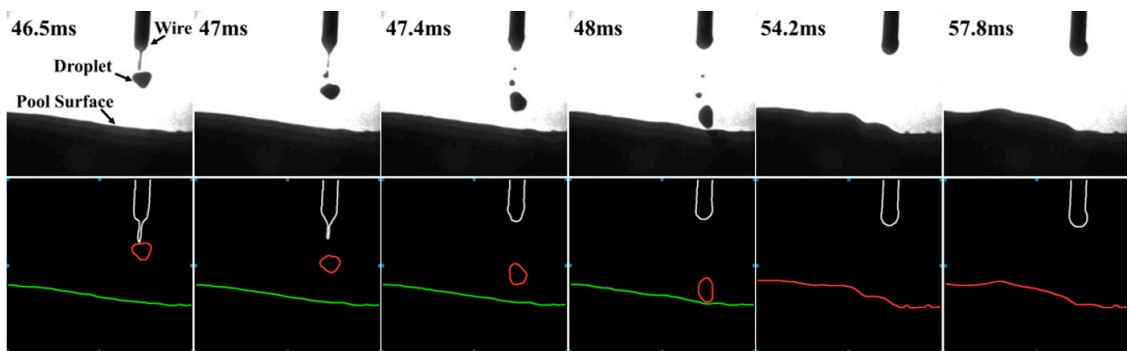

**Figure 3.** Contour extraction of droplets and weld pool.

A reference point (A) was defined on the weld pool surface to trace the pool surface. The x coordinate of the reference point is constant; the fluctuation of the y coordinate of the reference point is the direct information about the weld pool oscillation. To avoid the hindrance of droplet transition to the surface contour extraction of the weld pool, the reference point (A) was located on the weld pool surface 1.8 mm from the center of welding wire, as shown in Figure 4.

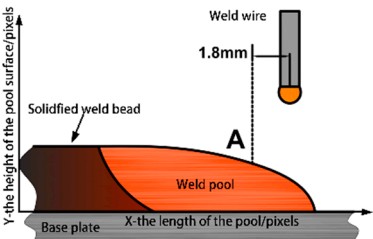

**Figure 4.** Position of the reference point: A.

The center of gravity of the droplet as approximated to the average value of its contour coordinate, as calculated by Equation (1) [1].

$$(x_{Gt}, y_{Gt}) = (\frac{\sum_1^n x_{nt}}{n}, \frac{\sum_1^n y_{nt}}{n}) \tag{1}$$

$(x_{Gt}, y_{Gt})$ is the coordinate of the center of droplet at t, $(x_n, y_n)$ are the coordinates of the droplet contour, n is the number of contour pixels. The droplet diameter can be calculated by Equation (2) [1].

$$D_{droplet} = \sqrt{(4 \times S_{droplet})/\pi} \tag{2}$$

$D_{droplet}$ is the equivalent diameter of droplets, $S_{droplet}$ is the area of droplet profile calculated by the number of pixels surrounded by the droplet contour line. Droplet velocity can be calculated by measuring the center coordinate of droplet in continuous photographs, as shown in Equation (3).

$$V_{droplet} = \frac{\sqrt{(x_{Gt1} - x_{Gt2})^2 + (y_{Gt1} - y_{Gt2})^2}}{|t_2 - t_1|} \tag{3}$$

$(x_{Gt1}, y_{Gt1})$ and $(x_{Gt2}, y_{Gt2})$ are the coordinates of the center of droplet at $t_1$ and $t_2$.

Changing the exposure time and the number of filters of high-speed camera can obtain different shooting effects. Increasing the exposure time and the number of filters equipped on the camera can cause the background light stronger than the arc light, which can filter the arc light, as shown in Figure 5a. Reducing exposure time and the number of filters leads to a higher arc light intensity than the background light intensity, and a precise arc contour can be obtained, as shown in Figure 5b. The arc characteristics were defined by its root diameter ($D_R$) and projected diameter ($D_P$) during pulse on the period, as schematically shown in Figure 5b.

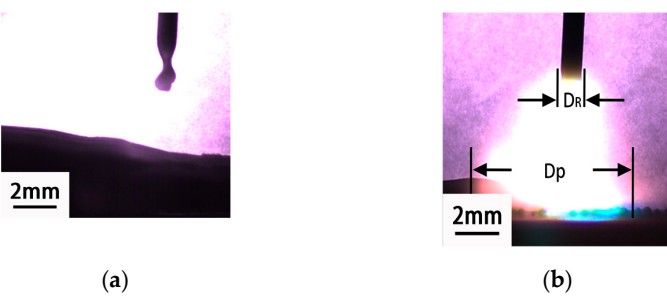

(a)                                              (b)

**Figure 5.** (**a**) Complete filtering out of the arc; (**b**) typical nature of arc profile.

### 2.3. Calculation of Cooling Rate, Growth Rate and Thermal Gradient

The cooling rate of weld can be evaluated by the two equations shown below [6].
For thick plate:

$$C_R = 2 \times \pi \times k \times (T_C - T_0)/H_{net} \tag{4}$$

For thin plate:

$$C_R = 2 \times \pi \times k \times \rho \times c \times t^2 \times (T_C - T_0)^3 / H_{net}^2 \tag{5}$$

where $C_R$ = cooling rate (K·s$^{-1}$), $k$ = thermal conductivity = 15 W·m$^{-1}$·K$^{-1}$, $\rho$ = density = 7850 Kg·m$^{-3}$, $c$ = specific heat = 500 J·Kg$^{-1}$·K$^{-1}$, $t$ = plate thickness(mm), $T_C$ = peak temperature = 1534.15 K, $T_0$ = final temperature = 300.15 K and $H_{net}$ = heat input rate(J·m$^{-1}$). The relative plate thickness factor was derived to select the proper cooling rate Equation [6]:

$$\tau = t \times \sqrt{\rho \times c \times (T_C - T_0)/H_{net}} \tag{6}$$

Equation (4) is applicable when $\tau \geq 0.75$, else Equation (5) is appropriate. In this study, 3 mm thin base plate was used, $\tau$ of all the studied conditions was lower than 0.75. Therefore, Equation (5) was selected to calculate the cooling rate. The dendrite growth rate in the solidification zone at the end of the weld is calculated as follows [6]:

$$R = v\cos\theta \tag{7}$$

where $R$ = growth rate (mm/s), $v$ = welding speed (mm/s) = 3.5 mm/s, $\theta$ = the angle between the normal to solidification front and the welding direction. The calculation equation of the thermal gradient of weld pool (G, K × mm$^{-1}$) without considering the convection of the weld pool is as follows [6]:

$$G = C_R/R \tag{8}$$

### 2.4. Sample Fabrication

A commercial 304 stainless steel plate of 200 mm × 150 mm × 3 mm was used as a base plate to prepare the bead on plate welds, using 308L stainless steel wire of 1.2 mm (nominal diameter) as the electrode. The weld groove shape is "I" with no gap. The chemical composition of the base plate and filler wire is given in Table 1. Gas composed of 98% argon and 2% $O_2$ was used as shield gas (20 L/min). The contact tip to base plate distance was 15 mm. In this paper, the influences of current waveform parameters on the weld pool behavior and microstructure of DP-GMAW were studied, which are thermal pulse frequency (TPF), thermal pulse current change ($\Delta I$) and duty ratio of thermal pulse peak phase ($D_{Tp}$), respectively. As a comparison, the weld pool behavior and weld metal microstructure of P-GMAW were also studied. Welding parameters are listed in Table 2.

**Table 1.** Material characteristics of the base plate and welding wire.

| Materials | C | Si | Mn | Cr | Ni | S | P | N | Mo |
|:---:|:---:|:---:|:---:|:---:|:---:|:---:|:---:|:---:|:---:|
| 304 | ≤0.08 | ≤1 | ≤2 | 18–20 | 8–10.5 | ≤0.03 | ≤0.03 | ≤0.1 | - |
| 316L | ≤0.03 | ≤1 | ≤2 | 16–18 | 10–14 | ≤0.03 | ≤0.045 | - | 2–3 |

**Table 2.** Welding parameters.

| No. | Process | $I_{Tp}$ (A) | $I_{Tb}$ (A) | TPF (Hz) | ΔI (A) | $D_{Tp}$ (%) | V | Speed (mm/s) | Penetration |
|-----|---------|--------------|--------------|----------|--------|--------------|------|--------------|-------------|
| 1 | DP | 130 | 90 | 0.5 | 40 | 50 | 22.5 | 20 | Full |
| 2 | DP | 130 | 90 | 1 | 40 | 50 | 22.5 | 20 | Full |
| 3 | DP | 130 | 90 | 2 | 40 | 50 | 22.5 | 20 | Full |
| 4 | DP | 130 | 90 | 3 | 40 | 50 | 22.5 | 20 | Full |
| 5 | DP | 130 | 90 | 2 | 40 | 20 | 22.5 | 20 | Full |
| 6 | DP | 130 | 90 | 2 | 40 | 35 | 22.5 | 20 | Full |
| 7 | DP | 130 | 90 | 2 | 40 | 70 | 22.5 | 20 | Full |
| 8 | DP | 130 | 105 | 2 | 25 | 50 | 22.5 | 20 | Full |
| 9 | DP | 130 | 75 | 2 | 55 | 50 | 22.5 | 20 | Full |
| 10 | DP | 130 | 60 | 2 | 70 | 50 | 22.5 | 20 | Full |
| 11 | P | 90 | | - | - | - | 22.5 | 20 | Full |
| 12 | P | 110 | | - | - | - | 22.5 | 20 | Full |
| 13 | P | 130 | | - | - | - | 22.5 | 20 | Full |

**Note:** $I_{Tp}$ = the average current at the peak of the thermal pulse; $I_{Tb}$ = the average current at the base of the thermal pulse; TPF = thermal pulse frequency; ΔI = $I_{Tp} - I_{Tb}$, $D_{Tp}$ = duty ratio of thermal pulse peak phase; V = voltage; Speed = weld speed; DP = DP-GMAW, P = P-GMAW.

## 3. Results and Discussion

### 3.1. Effect of Arc and Droplet Transfer on Weld Pool

During the welding process, the arc pressure and the impingement of droplet agitate weld pool intensify the convection in the weld pool. Hence, the study on heat and mass transfer process definitely is the premise of that on the dynamic behavior of pool in DP-GMAW. In P-GMAW, the current pulse frequency is constant. In DP-GMAW, the frequency of the current pulse changes periodically. Typical electrical signal waveforms of P-GMAW and DP-GMAW.

A thermal pulse cycle of DP-GMAW consists of peak period, base period and transition period of thermal pulse, as shown in Figure 6. The peak and base period of thermal pulse differ in the frequency and the peak of current pulse, leading to variational heat input, arc pressure and droplet impingement force. Therefore, it is necessary to analyze the heat input and force acting on the weld pool during a single current pulse period, then analyze the variational process of the heat input and the force acting on the pool during the whole thermal period of DP-GMAW. The thermal nature of the DP-GMAW weld pool is largely controlled by the heat content of the droplet and the arc heating. Assuming that the current distribution is homogeneous on the arc projection plane on the pool surface, the arc heat during one current pulse ($E_{arc}$) is expressed as follows [16]:

$$E_{arc} = \int_0^{1/f} I(V_w - \varphi)dt \tag{9}$$

where $V_w$ is the cathode voltage when the cathode material is stainless steel and $\varphi$ is the electronic work function of stainless steel. Y Yokomizu [17] points out that for stainless steel, $V_w$ is about 16.7 v, $\varphi$ is 4.77 v. $P_{arc}$ is the instantaneous power of heat input of arc to weld pool. The heat content of the droplet ($E_{droplet}$) can be estimated by the following Equation [14]:

$$E_{droplet} = \rho h \frac{4}{3}\pi\left(\frac{D_d}{2}\right)^3 \tag{10}$$

where $\rho$ is the density of the liquid stainless steel, $h$ is the enthalpy of the droplet and $D_d$ is the diameter of the droplet. Considering overheating of the droplets, the assumed temperature before the droplets enter the molten pool is 2900 K, and the enthalpy of the droplets $h$ [16] is:

$$h = \int_{300}^{2900} C_p dT \tag{11}$$

where $C_p$ is the specific heat capacity of the droplet and T is the temperature. According to the research data of Davim [18], h = 1.578 × 106 J/kg. The total heat input ($E_{total}$) and the power of heat input ($P_{total}$) acting on weld pool in a single current pulse period are:

$$E_{total} = \rho h \frac{4}{3} \pi \left( \frac{D_d}{2} \right)^3 + \int_0^{1/f} I(V_w - \varphi) dt \tag{12}$$

$$P_{total} = E_{total} \times f \tag{13}$$

where $f$ is the instantaneous frequency of the current pulse. The arc force ($F_{arc}$) during welding process is as follows [18]:

$$F_{arc} = \frac{\mu}{4\pi} I^2 \log \frac{D_d}{D_R} \tag{14}$$

where $\mu$ is the space permeability of the arc area, $\mu = 4.073 \times 10^{-4}$ N/A$^2$ [19]. The momentum of the droplet ($p_{droplet}$) and the droplet impingement pressure on weld pool ($P_d$) are shown as follows [16]:

$$p_{droplet} = \frac{4}{3} \pi \left( \frac{D_{droplet}}{2} \right)^3 \rho \, V_d \tag{15}$$

$$P_d = \frac{2 f \rho D_d V_d}{3} \tag{16}$$

where $V_d$ is the velocity of the droplet. Combined with the above formula, the arc behavior, droplet transition behavior, heat input and force acting on the weld pool during a current pulse period were analyzed, as shown in Figures 7 and 8. The arc size increases first and then decrease with the change of current during a current pulse, as shown in Figure 8a. At the beginning of the current pulse, the current and voltage rise rapidly to the peak, the size of the arc, the pressure and the heat power of the arc acting on weld pool rises synchronously. During peak time, current, arc size, arc pressure rise to the maximum in the pulse period. The arc transmits most of the heat to the weld pool during peak time, as shown in Figure 8b. With the decrease of current, the size of arc decreases gradually. The arc pressure and the heating power also decreases synchronously with current. While the size of the arc increased after the time of the droplet detached from the wire. It was caused by metal vapor concentration increasing in arc space when the droplets detach from the wire, and the phenomenon of arc jumping is also one of the factors, as shown in Figure 7(5,6). While this increase of arc size has no obvious effect on arc force and heating process at low current. The heat contained by the droplet and its impact force acting on weld pool can be calculated by Equations (5) and (11). Based on the above data, the total heat input, the arc pressure and the droplet impingement force acting on the weld pool in a single pulse period can be calculated, as shown in Figure 8b.

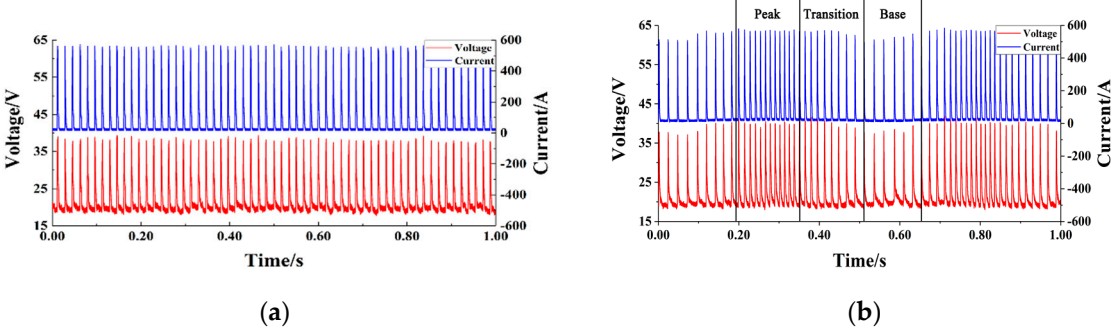

**Figure 6.** Welding electrical signal waveform of (**a**) P-GMAW(No.12); (**b**) DP-GMAW(No.3).

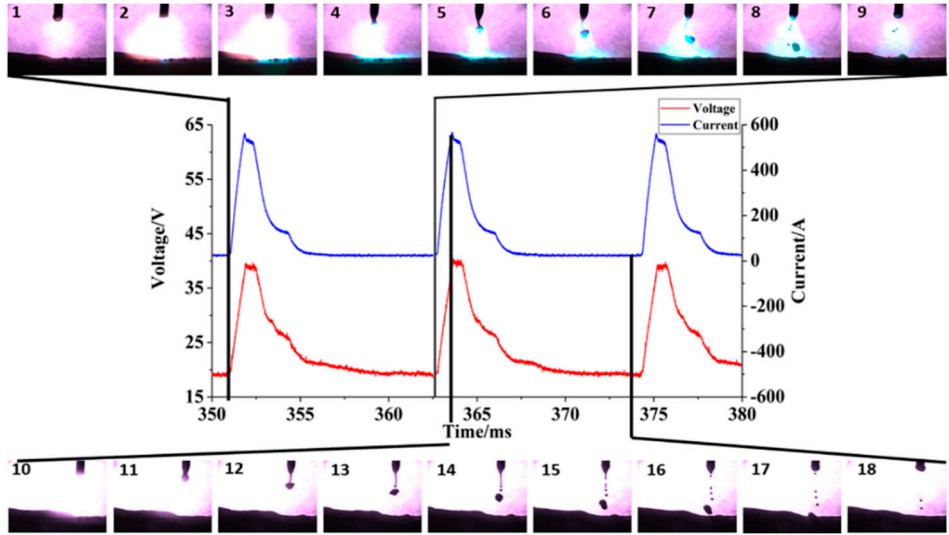

**Figure 7.** Combination of arc behavior, droplet transition behavior and electrical signals: 1–9 are the arc profile, 10–18 are images of the droplet transfer process (No.3).

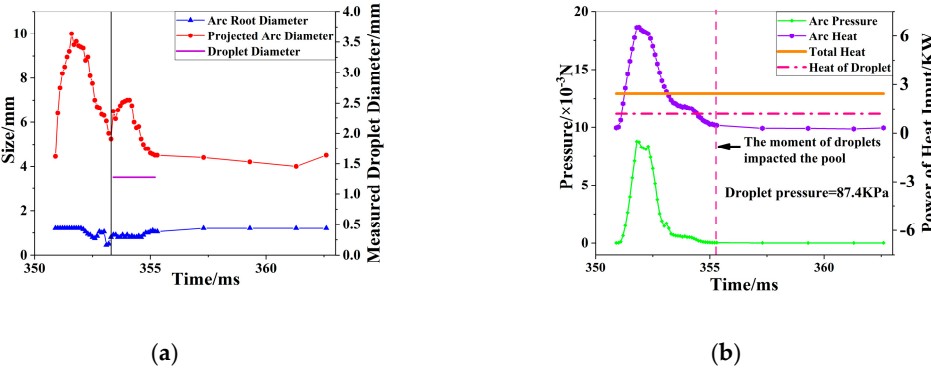

(**a**)                                                            (**b**)

**Figure 8.** (**a**) Arc size and droplet diameter; (**b**) heat input and pressure acting on weld pool.

DP-GMAW process periodically adjusts the output welding current, pulse waveform and pulse frequency vary with the current on the basis of "one droplet one pulse". Figure 9 shows the peak pulse current and pulse frequency with different output welding current. With the increase of the output current, the pulse frequency increases significantly, while the peak current of the pulse increases slightly. The main way to control output current for DP-GMAW process is to adjust the current pulse frequency. The purpose of adjusting the peak of the current pulse is to maintain the stability of the transition under different welding currents.

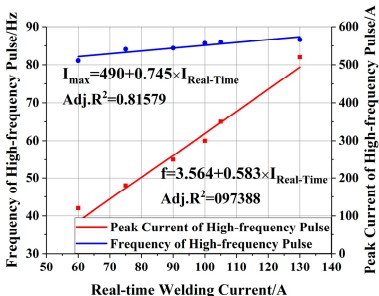

**Figure 9.** Peak current and frequency of pulses with different welding currents.

Figure 10 shows the heat content, the power of heat input, the pressure of arc, the droplet impingement force and momentum per unit current pulse with different output welding current. As shown in Figure 10a, the heat content of droplet was similar under different output welding current the arc heat had the same trend. Due to the uneven composition of the welding wire and the unstable wire feed speed during welding process, the size of the droplets cannot be consistent. However, the difference in droplet size with different average currents were small due to the consistency of the current pulse waveform. However, the heating power acting on weld pool increased obviously due to a clear growth of current pulse frequency. The momentum carried by the droplets were similar under different output welding currents, as shown in Figure 10b. Many studies can contribute to explaining it. The research of Emanuel [1]. Found that the droplet speed depends on the ratio between base to peak current of P-GMAW, which is different from the traditional GMAW. A slight change in the peak current could not significantly affect the droplet velocity. P.K. Ghosh's study [18] came to a similar conclusion that the diameter and speed of droplet in P-GMAW process predominantly depends upon $I_p$ irrespective of mean current and arc voltage. The increase of droplet transition frequency leads to an obvious increase of equivalent droplet impact force, as shown in Figure 10b. It is worth noticing that the arc pressure acting on the weld pool was much less than the droplet impingement force. Other researchers have described similar phenomena that liquid waves in P-GMA welding are triggered primarily by the impact of droplet, not by arc pressure [20]. Therefore, this paper only considers the droplet impingement force on weld pool.

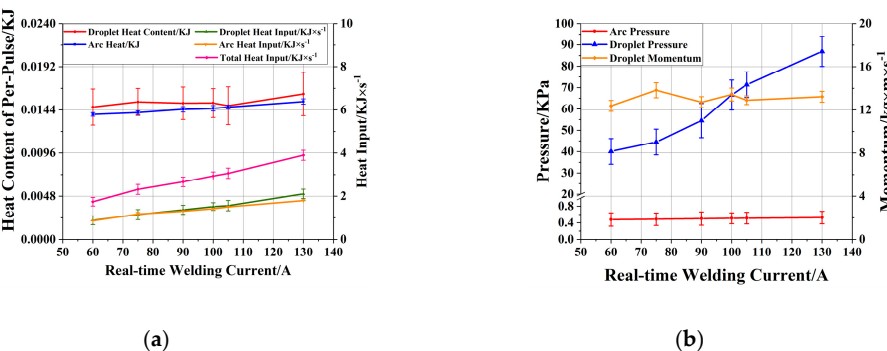

**Figure 10.** Heat input (**a**), pressure and momentum (**b**) of arc and droplet during single current-pulse with different welding currents.

Figure 11 shows the heat input and the droplet impingement force acting on the weld pool during the thermal pulse period with different ΔI. As shown in Figure 11, the process of thermal and pressure acting on the weld pool demonstrated the characteristic periodically fluctuation same with the thermal frequency. The heat input rate of the different stage of the thermal pulse were the average heat input rate during $T_p$ or $T_b$. This paper compares the heat input rate and the thermal gradient of weld pool (G) at different period under all parameters, as shown in Table 3. It should be noted that the thermal gradient of weld pool (G) along the welding direction calculated by Equation (7) is without considering the pool convection.

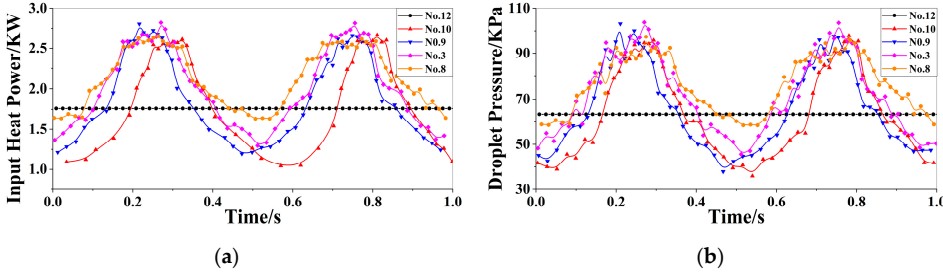

**Figure 11.** Changes of heat input (**a**) and droplet impingement force (**b**) in thermal pulse of DP-GMAW.

**Table 3.** Heat input rate (KJ/m) and the thermal gradient of weld pool (K × mm$^{-1}$) with different parameters.

| No. | 1 | 2 | 3 | 4 | 5 | 6 | 7 | 8 | 9 | 10 | 11 | 12 | 13 |
|---|---|---|---|---|---|---|---|---|---|---|---|---|---|
| $T_P$ | 68.2 | 72.5 | 66.2 | 68.7 | 71.5 | 66.7 | 71.8 | 82 | 60.6 | 52.1 | - | - | - |
| $T_b$ | 137 | 134.9 | 136 | 133.8 | 135.6 | 133.8 | 135.7 | 133 | 133 | 133 | - | - | - |
| Avg. | 102.6 | 103.7 | 101 | 101.2 | 84.3 | 90.2 | 116.5 | 107 | 96.8 | 92.6 | 71 | 102.2 | 142 |
| $G_{Tp}$ | 67.3 | 59.6 | 71.4 | 66.3 | 61.2 | 70.4 | 60.7 | 46.5 | 85.2 | 115.2 | 62 | 30.4 | 15.5 |
| $G_{Tb}$ | 16.7 | 17.2 | 16.9 | 17.5 | 17.0 | 17.5 | 17.0 | 17.8 | 17.7 | 17.6 | | | |

### 3.2. The Behavior Characteristics of Weld Pool in Double-Pulsed GMAW

In the process of DP-GMAW, the thermal and pressure acting on the weld pool of $T_P$ and $T_B$ are significantly different, the dynamic behavior of weld pool were varied with thermal frequency (F). For a better understanding of the influence of DP-GMAW current waveform parameters on the dynamic behavior of austenite stainless steel weld pool, the profile and oscillation characteristics under different current waveform parameters were recorded and extracted. In order to simplify the analysis process, two typical current waveform parameters (No.3 and No.12) were selected to summarize the weld pool profile.

The heat and the mass transferred to the weld pool during the period of a single current pulse can hardly affect the shape of weld pool significantly. In P-GMAW, the heat input and the droplet impingement force acting on weld pool are constant during welding process, and the profile of the weld pool caused by them remains stable, as shown in Figure 12a. In DP-GMAW, the length and the width of weld pool in the base period of thermal phase were obviously smaller than that in the peak period, as shown in Figure 12b,c. The smaller heat input and less metal deposition lead to rapid shrinkage of pool size during base period. The pool trailing edge shrank and separated from the solidified bead boundary of the weld, as shown in Figure 12c. This is the main factor that forms bead surface ripple. The shrinkage and expansion of the weld pool outline is mainly affected by the fluctuation of heat transfer and mass transfer in $T_p$ and $T_b$ period of thermal pulse.

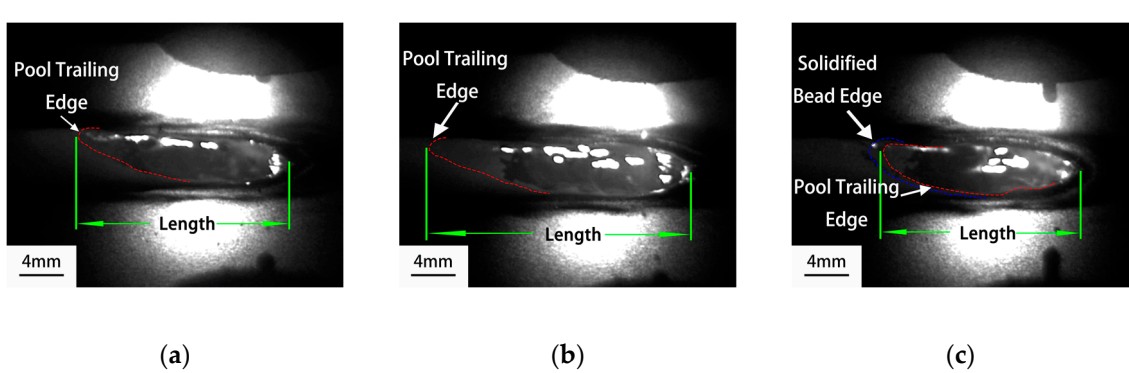

(**a**)　　　　　　　　　　　　　　　(**b**)　　　　　　　　　　　　　　　(**c**)

**Figure 12.** The variation of weld pool profile in thermal pulse: (**a**) P-GMAW (No.12); (**b**) peak period of thermal pulse of DP-GMAW (No.3); (**c**) Base period of thermal pulse of DP-GMAW (No.3).

The length variation of the weld pool under different parameters of the current waveform are shown in Figure 13. Figure 13a shows the weld pool length at $T_p$ and $T_b$ with different TPF. During the P-GMAW welding process, the weld pool can be regarded as a constant. The blue and red curves of Figure 13a are the curve of weld pool length at $T_p$ and $T_b$, respectively, with the thermal frequency. The descending trend of blue curve and the ascending red curve indicates that pool length at $T_p$ is gradually shortened while that at Tb is gradually increased with the increase of heat pulse frequency. It indicates that with the increase of thermal frequency, the period of $T_p$ decreases, and the heat accumulation and droplet transition acting on the weld pool gradually decrease, resulting in the decrease of the weld pool length at $T_p$. While the differences of the heat and droplet transfer

amount into the weld pool at $T_p$ and $T_b$ stage gradually decreases, the pool length difference between $T_p$ and $T_b$ gradually decreases but both close to the length of P-GMAW.

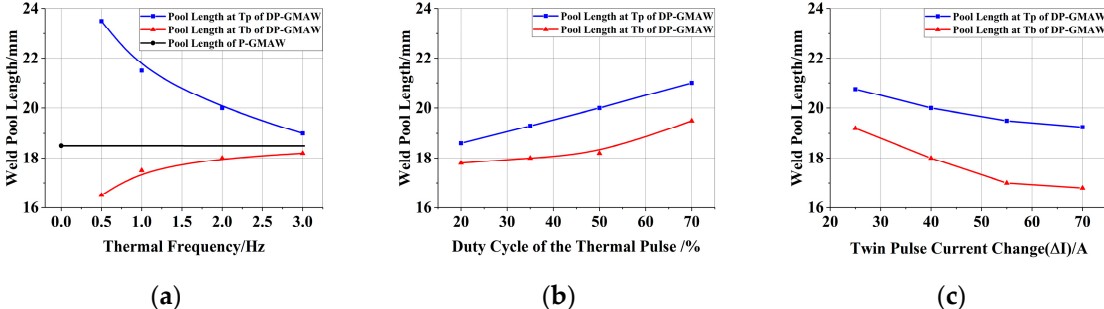

**Figure 13.** Variation of weld pool length with (**a**) heat thermal frequency; (**b**) duty cycle of the thermal pulse; (**c**) current amplitude of thermal pulse.

Figure 13b shows the pool length at $T_p$ and $T_b$ with different $D_{Tp}$. With larger $D_{Tp}$, the total heat input and droplet transition acting on the weld pool increase, the pool length at $T_p$ and $T_b$ both increasing. It has to be noticed that as $D_{Tp}$ increases, the difference between the pool length at $T_p$ and $T_b$ always increases first and then decreases. Small $D_{Tp}$ could cause longer $T_b$ with fixed thermal pulse frequency, the heat accumulation in $T_p$ was too small to significantly expand the size of weld pool. A larger $D_{Tp}$ could cause smaller $T_b$, shorter low heat input time ($T_b$) could not cause a significant reduction in the size of the pool. Too large or too small $D_{Tp}$ could result in small length difference.

Figure 13c shows the pool length at $T_p$ and $T_b$ with different $\Delta I$. With larger $\Delta I$, the total heat input and droplet transition acting on the weld pool decrease, the pool length at $T_p$ and $T_b$ both decreasing. Larger $\Delta I$ increases the differences of heat accumulation and amount of deposited metal between the $T_p$ and $T_b$.

The oscillation process of weld pool was recorded by the method mentioned in Section 2.2. Two typical current waveform parameters (No. 3 and No. 11) were selected to summarize the oscillation process of the weld pool, as shown in Figure 14. The fluctuation in the amplitude of the pool oscillation during the P-GMAW welding process was constant (Figure 14a), while that during the DP-GMAW was obvious (Figure 14b). That is the oscillation amplitude of weld pool in $T_p$ is much larger than that in $T_b$, since the droplet impingement force of P-GMAW and the size of weld pool were stable (Figures 11b and 13a), the amplitude of pool oscillation can be regarded as a constant.

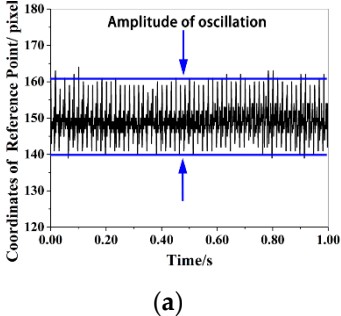
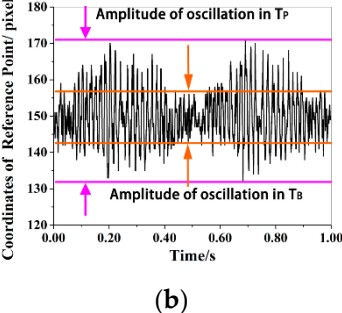

**Figure 14.** Height of reference point A as a function of time: (**a**) P-GMAW(No.12); (**b**) DP-GMAW(No.3).

In DP-GMAW, the fluctuation process of oscillation amplitude is similar to the process of thermal and pressure on the weld pool, as shown in Figure 11. The droplet impingement force on the weld pool increases significantly when the high-frequency current pulses increase the volume of weld pool during $T_p$ period, which is the main reason for the larger oscillation amplitude of the weld pool. The size of the weld pool and the droplet impingement force simultaneously resulting in a decrease

in the amplitude of oscillation during Tb period. The variation of oscillation amplitude of weld pool under different parameters of the current waveform are shown in Figure 15.

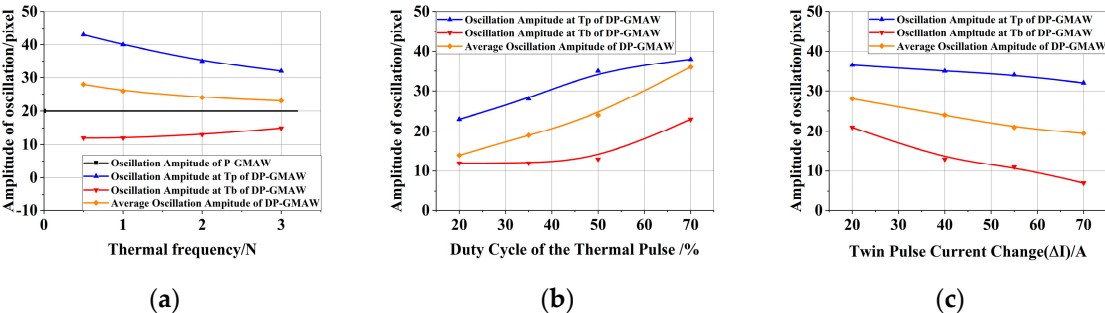

**Figure 15.** Variation of weld pool oscillation amplitude with (**a**) heat thermal frequency; (**b**) duty cycle of the thermal pulse; (**c**) current amplitude of thermal pulse.

The curves of the weld pool amplitude in $T_p$ and $T_b$ under different thermal pulse frequencies had a similar trend to the weld pool length, as shown in Figures 13a and 15a. Increasing the frequency of thermal pulses led to a reduction in the difference between the size of the weld pool at $T_p$ and $T_b$. As for the oscillation amplitude at $T_p$, the decrease in the size of the weld pool can decrease the oscillation amplitude under the same droplet impingement force. The increase in the weld pool size at $T_b$ can increase the oscillation amplitude, as shown in the red line of Figure 15a. However, there is no denying the fact that the average oscillation amplitude of DP-GMAW was greater than that of P-GMAW.

Larger duty cycle of the thermal pulse ($D_{Tp}$) can trigger greater oscillation amplitude with larger pool size and longer high-frequency droplet impingement. too large or too small $D_{Tp}$ could result in small amplitude difference of the oscillation amplitude within $T_p$ and $T_d$, as shown in Figure 15b.

Larger $\Delta I$ will increase the difference between the pool size and the droplet impingement force in $T_p$ and $T_b$, leading to a greater oscillation amplitude difference. However, the decrease of total heat input decreased the average amplitude of weld pool, as shown in Figure 15c.

So, as to what is known, DP-GMAW weld pool behavior is more complicated compared with P-GMAW. During switching from $T_p$ to $T_b$, the pool size experiences "expanding–shrinking" variation, the change of the oscillation amplitude of weld pool is synchronized with the pool size.

### 3.3. Effect of Process Parameters on Microstructures

The microstructure of fusion weld metal is controlled by the solidification behavior of weld pool [21]. Metallographic observation of all weld cross-sections with different welding parameters was conducted, and no defects such as incomplete fusion and porosity can be found. As mentioned above, the thermal pulse of DP-GMAW caused significant fluctuations in weld pool size and oscillation amplitude during welding process. It resulted in significant differences in welding microstructure between P-GMAW and DP-GMAW. The weld cross-sections, typical microstructures and heat affected zone (HAZ) of different weld processes (No.3 and No.12) are shown in Figures 16 and 17.

The microstructure of the P-GMAW weld was composed of coarse austenite ($\gamma$) columnar structures and small part of equiaxed austenite grain. The equiaxed crystal regions are distributed in the gaps between the ends of the columnar crystal regions, there is no obvious boundary between the equiaxed crystal regions and the columnar crystal regions, as shown in Figure 16b. The ferrite ($\delta$) morphology was skeleton-shaped, distributed in the columnar austenite grain gap.

Compared to P-GMAW, the microstructure of DP-GMAW weld exhibited an obvious refinement of microstructure along with large distribution of equiaxed crystal regions. A clear boundary emerged between the equiaxed crystal area and columnar crystal area, as shown in Figure 17b. Although the structure of austenite columnar crystals was significantly refined, it can be found that the ferrite

size increased significantly by comparing the ferrite morphology in austenite gap between P-GMAW and DP-GMAW, as shown in Figures 16d and 17d.

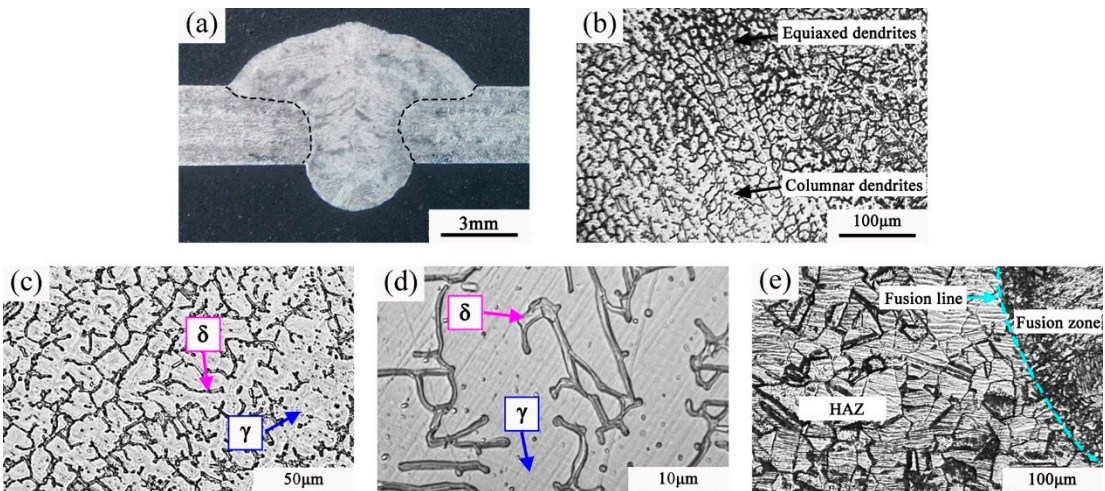

**Figure 16.** Optical micrographs of microstructure and heat affected zone (HAZ) of P-GMAW (No.12): (**a**) weld cross-section; (**b**) typical microstructure of P-GMAW weld, (**c**) coarse austenite columnar structures, (**d**) ferrite morphology and (**e**) HAZ.

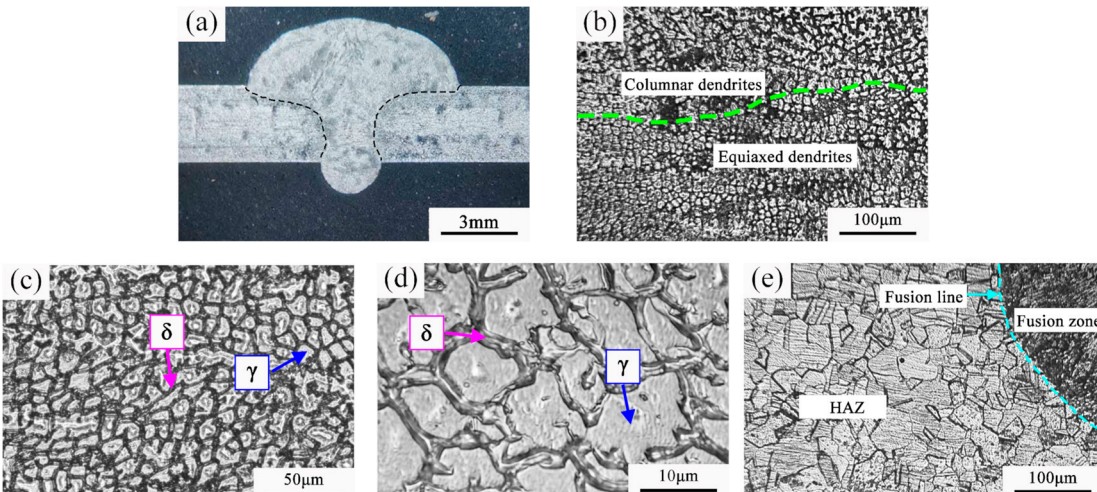

**Figure 17.** Optical micrographs of microstructure and HAZ of DP-GMAW (No.3): (**a**) weld cross-section; (**b**) typical microstructure of DP-GMAW weld, (**c**) austenite equiaxed dendrites structures, (**d**) ferrite morphology and (**e**) HAZ.

Figure 18 is a schematic sketch explaining how the thermal pulse of DP-GMAW helps grain refining. The dynamic behavior of the weld pool and the fluctuation of solidification parameters are the main factors that trigger the refinement of structure of DP-GMAW. The thermal pulse of DP-GMAW caused synchronized periodical fluctuations in heat input and pool amplitude, as shown in Figure 18a. Low heat input led to the large temperature gradient G of the weld pool during $T_b$. Ferrite was the primary grain in weld pool, ferrite columnar dendrites dominate with less constitutional supercooling (the area surrounded by $T_L$ and $T_{actual}$), as shown in Figure 18b.

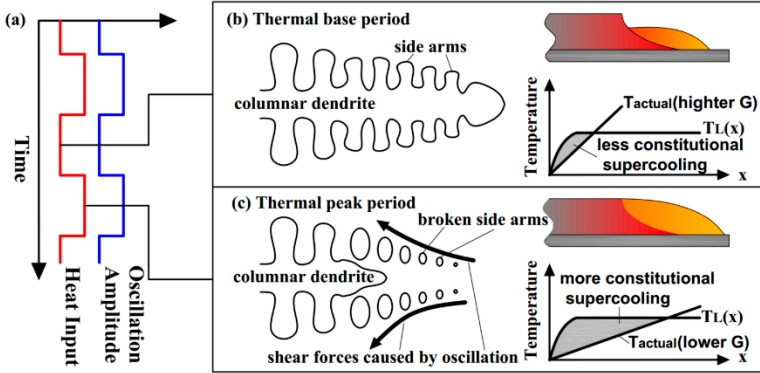

**Figure 18.** Thermal pulse helping grain refining: (**a**) process of the heat input and pool oscillation amplitude in DP-GMAW; (**b**) constitution supercooling in thermal base period; (**c**) constitution supercooling in thermal peak period.

The high heat input during $T_p$ can reduce the temperature gradient G in weld pool. The oscillation amplitude of weld pool was greatly increased, resulting in the enhancement of convection, which is expected to reduce the temperature gradient G in weld pool, leading to more constitutional supercooling (the area surrounded by $T_L$ and $T_{actual}$) in the weld pool, as shown in Figure 18c. The peak thermal period can cause reheating and melting of dendrite arms, thus hindering the further crystal growth and causing dendrite fragmentation [22]. At the same time, the shear stress produced by the enhanced convection in weld pool aggravates the dendrite fragmentation during $T_p$. The broken dendrite particles provide the necessary crystal nucleus for the liquid metal crystallization. Excessively constitutional supercooling was beneficial to the dendrite fragments survival and grow into equiaxed grains.

The equiaxed ferrite grains in the freshly solidified weld transforms into austenite by diffusion transformation in subsequent $T_b$ of thermal pulse. While the fast weld cooling speed reduced transition time of "$\delta \rightarrow \gamma$", both diffusion of ferrite-forming elements and austenite-forming elements were suppressed. It is the main factor leading to increasing the size and the content of ferrite structure in DP-GMAW weld microstructure compared to P-GMAW, as shown in Figure 16c,d and Figure 17c,d.

The microstructure of HAZ of weld joints prepared by P-GMAW and DP-GMAW are shown in Figures 16e and 17e. There is no obvious difference in the grain size of HAZ between the weld joints prepared by P-GMAW and DP-GMAW respectively. Similar heat input rate between welding parameters of No.11 and No.3 may be the main reason. The average size of weld microstructure and HAZ with different welding parameters were measured using the intercept method (as per ASTM E112-10), the values are statistic presented in Figure 19a. When the base metal of HAZ is heated, the microstructure undergoes a process of recrystallization, new undistorted equiaxed grains appear in the microstructure and gradually replace distorted grains. After the recrystallization is completed, continue to heat up or prolong the elevated temperature holding time could make the grain continue to grow. The size of HAZ in weld joint mainly depends on the heat input rate, Large heat input rate leads to longer elevated temperature holding time of HAZ, which is conducive to the diffusion and recrystallization process of tissue, thus leading to serious growth of grains. The fitting curve between grain size of HAZ and heat input rate is shown in Figure 19b. It can be found that the heat input rate has a good linear relationship with the grain size of HAZ, while no obvious correlation could be found between the grain size of HAZ and the thermal pulse of DP-GMAW.

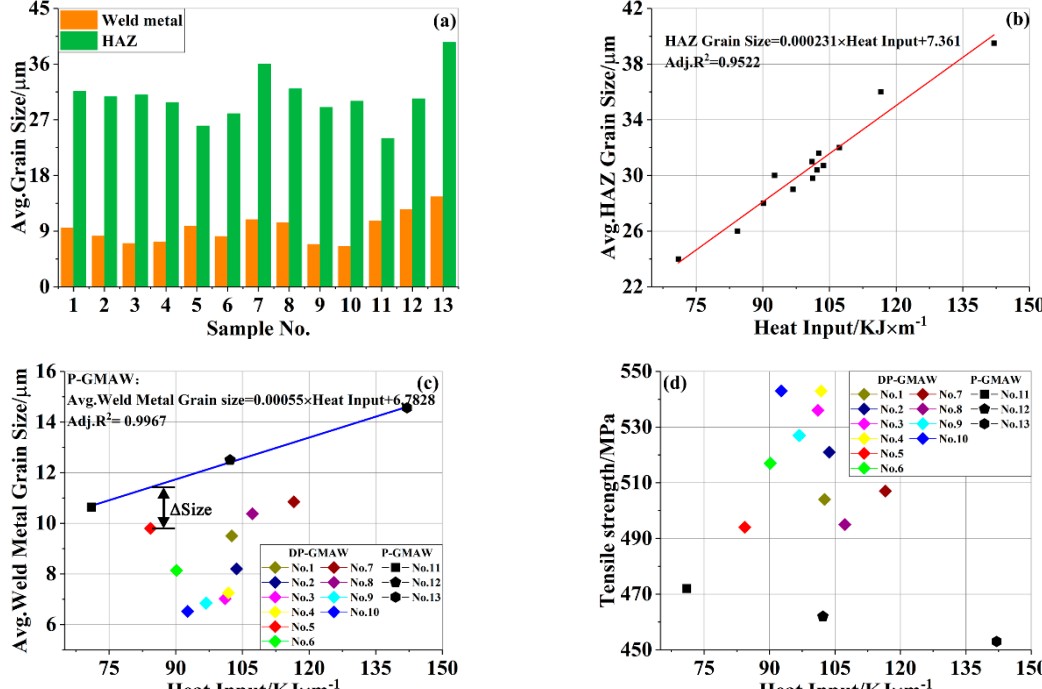

**Figure 19.** (**a**) Average grain size of weld metal and HAZ; (**b**) grain size of HAZ with different heat input; (**c**) grain size of weld metal with different heat input; (**d**) tensile strength with different heat input.

The increase of heat input rate could also coarsen the weld metal grain. The weld grain size has a good linear relationship with the heat input rate in P-GMAW [6,23], as shown in Figure 19c. In order to evaluate the beneficial effect of thermal pulse of DP-GMAW on grain refinement of weld metal microstructure with different welding parameters. "ΔSize" is used to characterize the degree of grain refinement, "ΔSize" is the difference between the size of DP-GMAW welding grain and P-GMAW welding grain under the same heat input. The grain size of P-GMAW welding was estimated by the linear fitting equation between the heat input and weld metal grain size of P-GAMW, as shown in Figure 19c. A larger "ΔSize" represents the more significant effect of thermal pulses on grain refinement. The transverse tensile test of the weld joints prepared by P-GMAW and DP-GMAW with different heat input is given in Figure 19d. It is observed that the tensile test of DP-GMA weld joint is higher than those of P-GMA weld joint due to the thermal pulse. As for the hardness of the weld joints, Ping Yao found that the variation characteristic of the hardness was approximately the same as that of grain size of weld joints of the 304 stainless steel prepared by P-GMAW and DP-GMAW [12].

Figure 20a shows "ΔSize" with different thermal pulse frequencies. In contrast, the average grain size of double pulse weld metal microstructure at thermal frequency = 2 was the smallest. Increasing the frequency of thermal pulses can effectively refine the grains of the weld metal with the given heat input rate. Figure 20b shows "ΔSize" with a different duty cycle of thermal pulse. A low duty cycle leads to the short duration of large constitutional supercooling of the weld pool, and the dendrite arms could not be melted off fully. Insufficient dendritic fragments can be the equiaxed nuclei in the weld pool. A large duty cycle leads high heat input which promote coarse-grain. Figure 20c shows "ΔSize" with different thermal pulse current change. Under the premise of maintaining the stability of the arc, the larger the current difference, the more effective the grain refinement of the thermal pulse. However, large current difference could result in poor weld formation. Therefore, increasing the current difference of thermal pulse cannot simultaneously obtain the fine grain and the good weld formation. It is necessary to analyze the actual situation in the application.

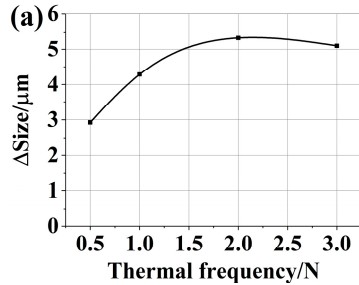 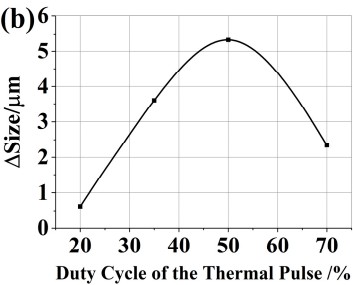 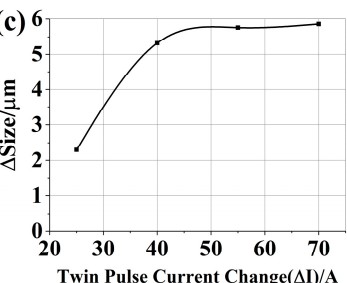

**Figure 20.** Variation of ΔSize with (**a**) heat thermal frequency; (**b**) duty cycle of the thermal pulse; (**c**) current amplitude of thermal pulse.

## 4. Conclusions

In this paper, a high-speed photography system and image processing technology were used to extract the characteristics of arc profile, the droplet transfer and the weld pool oscillation. The influence of DP-GMAW welding parameters on heat input, pressure acting on weld pool, weld pool size and oscillation amplitude have been calculated. Additionally, the internal relation between weld pool behavior and microstructure was analyzed. The conclusions are as follows:

(1) In contrast with P-GMAW, the length and the oscillation amplitude of the weld pool show periodic changes within one thermal pulse of DP-GMAW. The thermal pulse led to remelting and resolidification of the weld bead near the pool trailing edge which shrank and separated from the solidified bead boundary of the weld during switching from $T_p$ to $T_b$.

(2) Welding pool oscillation caused by the thermal pulse enhances the weld pool convection, which can help dendrite fragmentation, thus providing sufficient crystal nucleus for liquid metal crystallizing. The convection can reduce the temperature gradient of the weld pool and increase constitutional supercooling of the weld pool to promote equiaxed grains surviving and growing.

(3) The size of HAZ in the weld joint mainly depends on the heat input rate. Thermal pulse of DP-GMAW has an insignificant effect on the grain size of HAZ.

**Author Contributions:** Methodology, T.C.; software, T.C.; investigation, T.C., P.Z. (Peng Zhang), writing—original draft preparation, T.C., P.Z. (Peng Zhang); writing—review and editing, S.X., T.C., B.W., P.Z. (Peizhuo Zhai); supervision, S.X.; project administration, S.X., W.L.; funding acquisition, S.X. All authors have read and agreed to the published version of the manuscript.

**Funding:** This work was funded by the National Natural Science Foundation of China, grant No.51675269 and the Priority Academic Program Development of Jiangsu Higher Education Institutions (PAPD).

**Acknowledgments:** The authors gratefully acknowledge the financial supports by the National Science Foundation of China under Grant numbers 51675269, as well as the Priority Academic Program Development of Jiangsu Higher Education Institutions (PAPD).

**Conflicts of Interest:** The authors declare no conflict of interest.

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
