# Peer review of "Investigation on the Dynamic Behavior of Weld Pool and Weld Microstructure during DP-GMAW for Austenitic Stainless Steel"

_metals, doi:10.3390/met10060754_

Round 1

Reviewer 1 Report

Overall interesting study, but there are some Major concerns/Questions:

please show some weld profiles and also macrosections of the welded joints to have a better comparison between the processes

when comparing 15 d) with 16 d) one can see totally different fusion lines; 15 d) is much deeper wheras 16 d) is shallower but broader

-> was that done with the same heat-input? otherwise you cannot compare those results

-> furtheron from a practical point of view you should compare welds with the same penetration e.g. 2 mm as a welder needs first of all the penetration to guarantee the strength of the joint

ll 364/365 please specify "low heat input" and especially "large temerature gradient" how huge are the differences? Are they really significant?

please specify "more constitutional undercooling"

how did you make sure that the compared heat inputs between P and DP were the same?

Does the smaller grainsize have any positive effects on the weld strength/ductility? (results from mechanical testings are expected)

Author Response

Dear reviewer,

We deeply appreciate the time and effort you have spent in reviewing our manuscript. Based on your comments and suggestions, we have made a modification on the manuscript and the revisions were highlighted in red in the revised manuscript. Here below are our descriptions of the revisions according to your comments.

Point 1: please show some weld profiles and also macrosections of the welded joints to have a better comparison between the processes.

Response 1: Thanks very much for your careful review of our manuscript. We have added the macro sections of the welded joints in Figure 16 and Figure 17.

Point 2:  when comparing 15 d) with 16 d) one can see totally different fusion lines; 15 d) is much deeper wheras 16 d) is shallower but broader

Response 2: Thanks very much for your careful review of our manuscript. It is your misunderstanding caused by our imprecision in the process of taking metallographic pictures. Both weld joints were full penetration. The difference in camera rotation angle results in a significant difference in the direction of the fusion line. We corrected the error and recaptured the picture, as shown in Figure 17(e).

Point 3: was that done with the same heat-input? otherwise, you cannot compare those results

Response 3: Thanks very much for your careful review of our manuscript. The heat input rates with all parameters were calculated, as shown in Table 3, and the heat input of No.3 and No.12 were similar. 

Point 4: further on from a practical point of view you should compare welds with the same penetration e.g. 2 mm as a welder needs first of all the penetration to guarantee the strength of the joint

Response 4: Thanks very much for your careful review of our manuscript. The weld joints with all parameters were all full penetration. We added this part in Table 2.

Point 5: 364/365 please specify "low heat input" and especially "large temperature gradient" how huge are the differences? Are they really significant?

Response 5: Thanks very much for your careful review of our manuscript. we added part 2.3 to explain the method for calculating temperature gradient without considering the convection of the weld pool. and the temperature gradient of the weld pool in Tp and Tb were presented in Table 3. The convection of the weld pool could further expand the difference between the temperature gradient of the weld pool in Tp and Tb.

Point 6: please specify "more constitutional undercooling"

Response 6: Thanks very much for your careful review of our manuscript. Degree of constitutional undercooling can be characterized by the area surrounded by TL and Tactual. Low temperature gradient could result in more constitutional supercooling, as shown in Figure 18(c).

Point 7: how did you make sure that the compared heat inputs between P and DP were the same?

Response 7: Thanks very much for your careful review of our manuscript. The heat input rates with all parameters were calculated, as shown in Table 3, and the heat input of No.3 and No.12 were similar. 

Point 8: Does the smaller grain size have any positive effects on the weld strength/ductility? (results from mechanical testings are expected)

Response 8: Thanks very much for your careful review of our manuscript. The result of the tensile strength of all weld joints was present in Figure19(d). It is observed that the tensile test of DP-GMA weld joint is higher than those of P-GMA weld joint due to thermal pulse.

Many thanks for your time and consideration.

Best regards

Tao Chen

taocmsc@nuaa.edu.cn

Reviewer 2 Report

The paper presents the investigation results related to the dynamic behaviour and structure modification of weld pool during DP-GMAW of austenitic stainless steel. The manuscript has potential, but there are many shortcomings, as follows:

  1. The title is too long and has to be rewritten.
  2. Confusion related to technical terms - specific to the welding field - has been noticed within the text and figures. There are many wrong technical terms which are inacceptable for a scientific article (Ex: weld system, welding pool, weld speed, heat input power, welding line energy, weld process, fusion welding metal, residence time, tissue etc.). I strongly recommend using the “Standard Welding Terms and Definitions” approved by International Welding Societies. Consequently, many paragraphs have to be revised and rewritten.
  3. Incorrect measure units have been found in the manuscript (Ex: heat input in [kJ/s], Kj). I strongly recommend consulting the “Standard Welding Terms and Definitions” approved by International Welding Societies.
  4. Lines 388-389: Which is the technical connection between the recrystallisation process and the TISSUE?
  5. The discussion related to results presented in figures, images, charts has to appear before them.
  6. The resolution of microstructures’ images is poor and has to be improved.
  7. Because the fonts’ size used within figures is inadequate, the information is mostly illegible.
  8. Translation into English language has to be revised by a native speaker who knows well technical terms, too.

Author Response

Dear reviewer,

We deeply appreciate the time and effort you have spent in reviewing our manuscript. Based on your comments and suggestions, we have made a modification on the manuscript and the revisions were highlighted in red in the revised manuscript. Here below are our descriptions of the revisions according to your comments.

Point 1: The title is too long and has to be rewritten.

Response 1: Thanks very much for your careful review of our manuscript.  The title was rewritten, and the new tile is "Investigation on the dynamic behavior of weld pool and weld microstructure during DP-GMAW for Austenitic stainless steel".

Point 2-3: Confusion related to technical terms - specific to the welding field - has been noticed within the text and figures. There are many wrong technical terms which are inacceptable for a scientific article (Ex: weld system, welding pool, weld speed, heat input power, welding line energy, weld process, fusion welding metal, residence time, tissue etc.). I strongly recommend using the “Standard Welding Terms and Definitions” approved by International Welding Societies. Consequently, many paragraphs have to be revised and rewritten. Incorrect measure units have been found in the manuscript (Ex: heat input in [kJ/s], Kj). I strongly recommend consulting the “Standard Welding Terms and Definitions” approved by International Welding Societies.

Response 2-3: Thanks very much for your careful review of our manuscript. We corrected inappropriate expressions in the manuscript in accordance with “Standard Welding Terms and Definitions -AWS A3.0MA3.0-2010 ”. 

Point 4: Lines 388-389: Which is the technical connection between the recrystallisation process and the TISSUE?

Response 4: Thanks very much for your careful review of our manuscript. When the base metal of HAZ is heated, the microstructure undergoes a process of recrystallization. new undistorted equiaxed grains appear in the microstructure and gradually replace distorted grains. After the recrystallization is completed, continue to heat up or prolong the elevated temperature holding time could make the grain continue to grow.

Point 5: The discussion related to results presented in figures, images, charts has to appear before them.

Response 5: Thanks very much for your careful review of our manuscript. Based on your suggestions, we have restructured the article and express our sincere thanks again.   

Point 6: The resolution of microstructures’ images is poor and has to be improved

Response 6: Thanks very much for your careful review of our manuscript. We have improved the image quality, thanks again.

Point 7: Because the fonts’ size used within figures is inadequate, the information is mostly illegible.

Response 7: Thanks very much for your careful review of our manuscript. We adjusted the fonts’ size used within figures。

Point 8: Translation into English language has to be revised by a native speaker who knows well technical terms, too.

Response 8: Thanks very much for your careful review of our manuscript. We have checked all spellings and grammatical mistakes carefully. We hope you will be satisfied with the revisions for the resubmitted manuscript. If you have any queries and suggestions, please do not hesitate to contact me.

We deeply appreciate the time and effort you have spent in reviewing our manuscript. Your comments and suggestions are all valuable and very helpful for revising and improving our paper. Once again, many thanks for your time and consideration.

Best regards

Tao Chen

taocmsc@nuaa.edu.cn

Reviewer 3 Report

This research was aimed to study the effect of heat and droplet transfer process on the dynamic behaviour of welding pool and weld microstructure during DP-GMAW for austenitic stainless steel. The subject under analysis is interesting and a relevant experimental work was conducted by the authors. 

This paper has potential to be considered for publication.

However, the following aspects must be addressed by the authors:

1) The English must be improved.

2) The figures have a very poor quality. Most of the figures with graphs have a very low resolution. In addition, Figure 7 is confusing and a different structure should be adopted.

3) The X and Y axes should be included in the schematics of Figure 4.

4) The meaning of all the parameters from each equation must be indicated (exactly with the same form as they are presented in the equations - Dd vs Ddroplet). The units of the parameters must be indicated.

5) The production of 3 samples by P-GMAW must be explicitly referred and explained in the text of the Experimental Procedure.

6) Were the graphs displayed in Figure 6 obtained experimentally? Were they obtained for which parameters (Figure 6a and 6b)?

7) The typos heat Etotal perpluse and Ptotal perpluse should be corrected.

8) References supporting all the equations must be presented.

9) Lines 202-204:

The authors refer “droplet impingement force”, but a droplet pressure is indicated in Figure 7c (not force). This occurs many times along the paper.

How do the authors explain this?

10) The evolution of some parameters presented in Figure 9 is referred to be constant. However, some of these parameters present fluctuations (for example, droplet heat content and droplet momentum). These fluctuations must be explained.

11) The results displayed in Table 3 must be better explained.

12) Lines 249 and 250:

The authors refer:

“In order to simplify the analysis process, two typical current waveform parameters (No 3 and No 11) were selected to summarize the molten pool profile.”

However, this is not matching the caption of Figure 11a.

13) The macrostructure of the weld cross-sections should be presented. Is there any type of defects in the welds? This should be referred.

14) The typo “highter” in Figure 17 should be corrected.

15) The mechanical properties of the welds were not studied. The authors should discuss the effect of the differences observed in weld microstructure on the weld mechanical properties. This discussion should be supported by experiments and/or by literature.

Author Response

Dear reviewer,

We deeply appreciate the time and effort you have spent in reviewing our manuscript. Based on your comments and suggestions, we have made a modification on the manuscript and the revisions were highlighted in red in the revised manuscript. Here below are our descriptions of the revisions according to your comments.

Point 1: The English must be improved.

Response 1: Thanks very much for your careful review of our manuscript. We have checked all spellings and grammatical mistakes carefully. We hope you will be satisfied with the revisions for the resubmitted manuscript. If you have any questions and suggestions, please do not hesitate to contact me.

Point 2: The figures have a very poor quality. Most of the figures with graphs have a very low resolution. In addition, Figure 7 is confusing and a different structure should be adopted.

Response 2: Thanks very much for your careful review of our manuscript. We improved the figures with graphs , and we re-drawed Figure 7.

Point 3: The X and Y axes should be included in the schematics of Figure 4.

Response 3: Thanks very much for your careful review of our manuscript. We added the X and Y axes in the schematics of Figure 4.

Point 4: The meaning of all the parameters from each equation must be indicated (exactly with the same form as they are presented in the equations - Dd vs Ddroplet). The units of the parameters must be indicated.

Response 4: Thanks very much for your careful review of our manuscript. We indicated all the parameters with the units of each equation.

Point 5: The production of 3 samples by P-GMAW must be explicitly referred and explained in the text of the Experimental Procedure

Response 5: Thanks very much for your careful review of our manuscript. The production of 3 samples by P-GMAW was referred and explained in Line 155-156.

Point 6: Were the graphs displayed in Figure 6 obtained experimentally? Were they obtained for which parameters (Figure 6a and 6b)?

Response 6: Thanks very much for your careful review of our manuscript. We add the welding parameters of Figure 6a and 6b.

Point 7: The typos heat Etotal perpluse and Ptotal perpluse should be corrected.

Response 7: Thanks very much for your careful review of our manuscript. The typos heat Etotal perpluse and Ptotal perpluse were corrected.

Point 8: References supporting all the equations must be presented.

Response 8: Thanks very much for your careful review of our manuscript. We added the references supporting all the equations.

Point 9: Lines 202-204:The authors refer “droplet impingement force”, but a droplet pressure is indicated in Figure 7c (not force). This occurs many times along the paper. How do the authors explain this?

Response 9: Thanks very much for your careful review of our manuscript. We corrected this mistake, and the "force" should be "pressure".

Point 10: The evolution of some parameters presented in Figure 9 is referred to be constant. However, some of these parameters present fluctuations (for example, droplet heat content and droplet momentum). These fluctuations must be explained.

Response 10: Thanks very much for your careful review of our manuscript. Due to the uneven composition of the welding wire and the unstable wire feed speed during welding process, the size of the droplets cannot be consistent. However, the difference in droplet size with different average currents were small due to the consistency of the current pulse waveform.

Point 11: The results displayed in Table 3 must be better explained.

Response 11: Thanks very much for your careful review of our manuscript. We added the explanation of Table 3 in Line 252-256.

Point 12: Lines 249 and 250:The authors refer:“In order to simplify the analysis process, two typical current waveform parameters (No 3 and No 11) were selected to summarize the molten pool profile.” However, this is not matching the caption of Figure 11a.

Response 12: Thanks very much for your careful review of our manuscript. We corrected this mistake, and the "No.11" should be "No.12".

Point 13: The macrostructure of the weld cross-sections should be presented. Is there any type of defects in the welds? This should be referred.

Response 13: Thanks very much for your careful review of our manuscript. We have added the weld cross-sections in Figure 16 and Figure 17. The explanation of the defects in the welds was added in Line 349-350.

Point 14: The typo “highter” in Figure 17 should be corrected.

Response 14: Thanks very much for your careful review of our manuscript. We corrected this mistake, as shown in Figure 18(c).

Point 15: The mechanical properties of the welds were not studied. The authors should discuss the effect of the differences observed in weld microstructure on the weld mechanical properties. This discussion should be supported by experiments and/or by literature.

Response 15: Thanks very much for your careful review of our manuscript. The result of the tensile strength of all weld joints was present in Figure19(d). It is observed that the tensile test of DP-GMA weld joint is higher than those of P-GMA weld joint due to thermal pulse.

We deeply appreciate the time and effort you have spent in reviewing our

manuscript. Your comments and suggestions are all valuable and very helpful 

for revising and improving our paper.

Once again, many thanks for your time and consideration.

Best regards

Tao Chen

taocmsc@nuaa.edu.cn

Round 2

Reviewer 1 Report

good revision

Author Response

Dear reviewer:

    Our deepest gratitude goes to the anonymous reviewers for your careful work and thoughtful suggestions that have helped improve this paper substantially.

                                                                                                 Tao Chen 

Reviewer 2 Report

Generally, the authors have modified the manuscript content, according to the reviewers’ comments. There are still shortcomings, as follows:

  1. Incorrect measure units have been found in the manuscript (Ex: heat input in [Kj∙cm-1]).
  2. The resolution of microstructures’ images is poor. Also, there are figures that have to be rebuilt.
  3. The fonts’ size used within figures is inadequate in comparison with the fonts’ size used within the text.

Author Response

Dear editor and reviewer

We have substantially revised our manuscript after reading your kind advice and

the comments provided by the reviewers. All the revisions have been highlighted

in the revised manuscript. We sincerely hope this manuscript will be finally

acceptable to be published on Metals. Thank you very much for all your

help.

Point 1:  Incorrect measure units have been found in the manuscript (Ex: heat input in [Kj∙cm-1]).

Response: Thanks for your precise comments. We are very sorry that our

revision did not make your satisfaction. Incorrect measure units (heat input in

[Kj∙cm-1]) have been corrected to“[Kj∙m-1]”.

Point 2: The resolution of microstructures’ images is poor. Also, there are

figures that have to be rebuilt.

Response: Thanks for your precise comments. We have resubmitted high-

resolution pictures and corrected the improperness in the pictures.

Point 3:The fonts’ size used within figures is inadequate in comparison with the

fonts’ size used within the text.

Response: Thanks for your precise comments. We have adjusted the size of

some fonts in the picture appropriately so that it has better clarity.

We deeply appreciate the time and effort you have spent in reviewing our

manuscript. Your comments and suggestions are all valuable and very helpful 

for revising and improving our paper.

Once again, many thanks for your time and consideration.

Best regards

Tao Chen

Reviewer 3 Report

Considering the changes conducted by the authors, the paper can now be accepted for publication.

Author Response

(The authors gave the same response as above.)
